# *Xenopus* Interferon Complex: Inscribing the Amphibiotic Adaption and Species-Specific Pathogenic Pressure in Vertebrate Evolution?

**DOI:** 10.3390/cells9010067

**Published:** 2019-12-26

**Authors:** Yun Tian, Jordan Jennings, Yuanying Gong, Yongming Sang

**Affiliations:** Department of Agricultural and Environmental Sciences, College of Agriculture, Tennessee State University, Nashville, TN 37209, USA; ytian@tnstate.edu (Y.T.); jjenni17@Tnstate.edu (J.J.); ygong@tnstate.edu (Y.G.)

**Keywords:** amphibian, *Xenopus*, interferons, antiviral, immune evolution

## Abstract

Several recent studies have revealed previously unknown complexity of the amphibian interferon (IFN) system. Being unique in vertebrate animals, amphibians not only conserve and multiply the fish-like intron-containing IFN genes, but also rapidly evolve amniote-like intronless IFN genes in each tested species. We postulate that the amphibian IFN system confers an essential model to study vertebrate immune evolution in molecular and functional diversity to cope with unprecedented pathophysiological requirement during terrestrial adaption. Studies so far have ascribed a potential role of these IFNs in immune regulation against intracellular pathogens, particularly viruses; however, many knowledge gaps remain elusive. Based on recent reports about IFN’s multifunctional properties in regulation of animal physiological and defense responses, we interpret that amphibian IFNs may evolve novel function pertinent to their superior molecular diversity. Such new function revealed by the emerging studies about antifungal and developmental regulation of amphibian IFNs will certainly promote our understanding of immune evolution in vertebrates to address current pathogenic threats causing amphibian decline.

## 1. The Unique Complexity and Role of Amphibian Interferons (IFNs) in Immune Evolution

The animal immune system relies on a series of specified immune molecules, collectively known as immunome, to detect and repel pathogenic invaders at the molecular, cellular, and organismal levels. Species-specific or cross-species characterization of the immunome lays a cornerstone in studying animal health and immune evolution [1,2,3]. Interferon (IFN)-mediated immune response involves IFNs and several hundreds of IFN-stimulated genes (ISGs) that play a pivotal role in antiviral defense [4,5,6,7]. Coincided with the evolution of adaptive immunity, IFN genes also originate in jawed fish from a few multi-exon progenitors [8]. Primitive IFN-like molecules have been previously determined in bony fish, with discernible IFN-γ- and IFN-ϕ-like genes, which are presumed to be progenitors of type II (IFN-II, IFN-γ) and type I IFNs (IFN-I), respectively, in tetrapods [9,10,11]. A new study discovered ancestral IFN orthologs of all three IFN types in a cartilaginous fish, and hence resolved that three types of IFNs were ramified before the bony fish [12]. We annotated IFN gene families across 110 vertebrate genomes, and showed that IFN genes, after originating in jawed fishes, had several significant evolutionary surges regarding the numbers of IFN-coding genes and subtypes in some species of amphibians, bats, and ungulates (particularly pigs and cattle of livestock) [13,14,15]. Here, we review the newly identified IFN molecular evolution and expansion in amphibians [12,13,16].

The evolution of intronless (or single-exon) IFN genes and further molecular diversification in amniotes are signature events in IFN evolution [11,12,13,14,15,16]. However, our understanding of this process is elusive, partly due to the unrecognized role of amphibians in IFN evolution [9,10,11]. Previous studies have: (1) Only discovered several intron-containing IFN progenitors in amphibians [9], and (2) underestimated IFN-mediated antimicrobial responses in amphibians and the unique role of amphibians in IFN evolution. Previously, the emergence of intronless IFN genes has been assumedly ascribed to reptiles, which further suggested a linear-increasing model for IFN diversification from fish to mammals [9,10]. Our recent research detected the unexpected emergence and expansion of intronless IFN genes in amphibians, exemplified by 40–50 predicated IFN-coding genes, of which 24–37 are intronless in each *Xenopus* species [13]. This remarkable coexistence of intron-containing and intronless IFNs in amphibians confirms that the emergence of intronless IFN genes actually occurred in amphibians (or even earlier in fish) in contrast to previously hypothesized in reptiles [13,16]. These findings determined previously unknown and species-independent expansion of intronless IFN genes in amphibians, which revises the established model of IFN evolution and highlights the critical role of amphibian IFN complex in studying IFN biology (Figure 1) [11,12,13,14,15,16].

For instance, extensive analysis of current genome assemblies of *Xenopus tropicalis* and *Xenopus laevis* at NCBI (https://www.ncbi.nlm.nih.gov/genome/?term=Xenopus) and Xenobase (http://www.xenbase.org/) led to the identification of 37 intronless IFN-like genes (XtIFNX) (including 4 near intronless IFN-like genes retaining only a short (<50 bp) intron) and 3 pseudogenes in *X. tropicalis* [13]. Similarly, 26 intronless IFN-I (XaIFNX) genes (including 8 with a short remaining intron and 4 pseudogenes) were identified in the *X. laevis* genome. Notably, those amphibian short-intron-containing IFN gene-like sequences were indicated to be rudimentary intermediates of the retro(trans)position event that further became intronless IFN genes. Our annotation of the current genome assemblies at NCBI and Xenbase not only determined intronless IFNs in frogs for the first time, but also discovered several more intron-containing IFNs, which were not revealed in previous studies. We therefore updated the current repertoire of IFN gene loci in *X. tropicalis*: The co-existence of 14 intron-containing IFNs (7 IFN-I, 1 IFN-II, and 6 IFN-III) and the expansion of 37 intronless IFN-like genes (36 IFN-I and 1 IFN-III); and in *X. laevis*: 17 intron-containing (7 IFN-I, 1 IFN-II, and 9 IFN-III) and 24 intronless IFN-coding genes (22 IFN-I and 2 IFN-III) (Table 1 and Figure 1) [13]. A new study in a Tibet frog species (*Nanorana parkeri*) reinforced the co-existence of both intron-containing and intronless IFN genes in amphibians [16]. However, the expansion of antiviral IFN genes in different amphibian species, especially the intronless ones, seems species-independent. These intronless amphibian IFN genes seem to have a paralogous rather than an orthologous relationship, as well as to the amniote intronless IFNs [12,13,16]. All of these studies collectively determined the unique coexistence of intron-containing and intronless IFNs in both IFN-I and IFN-III in amphibians [12,13,14,15,16]. These findings support that the retroposition process, which initiated the evolution and expansion of intronless IFNs in higher amniotes, evidently originated in amphibians or earlier, rather than in reptiles as previously believed [9,10,11,12,13,16]. The species-independent expansion of intronless type I IFNs in amphibians suggests a unique role of amphibians in IFN evolution. Indeed, collected evidence also indicates an evolutionary benefit of these intronless IFN genes in gene expression and efficient contest to infectious and developmental pressures during adaption to terrestrial lives [11,12,13,14,16,18,19,20].

Amphibians are an irreplaceable link in animal evolution toward terrestrial tetrapods. They reside in aquatic, terrestrial, and arboreal habitats, and are sensitive indicators for monitoring the environmental and habitat changes in our bio-ecosystem. In basic science research, amphibians have long been utilized as prominent models for studying physiological and developmental processes [21,22,23,24]. Due to the crucial role that amphibians play in both the ecosystem and experimental sciences, it is imperative to gain a better understanding of their immunobiology. Recent investigations have indicated that pathogenic factors (such as by chytrid fungus and Ranavirus) are causing amphibian population decline and threaten global biodiversity [25,26,27,28,29,30,31]. This further emphasizes the need to study the amphibian immunome and its interaction with epidemic pathogens [1,2,3,32,33,34,35,36,37,38,39,40]. Amphibians have several unusual characteristics that are stimulating for immunobiological studies [32,33,35,39]. For example, amphibians have moist and permeable skin, which also serves as an organ for respiration [33,34]. Previous studies have discovered that amphibian skin is one of the most generous sources of antimicrobial peptides, of which some are broadly effective and potentially therapeutic against bacteria, fungi, viruses, and even cancers [37,38,41,42,43,44,45]. Recent studies in the amphibian IFN system uncovered another unique advantage—that amphibians give rise to the emergence and expansion of intronless IFN genes [12,13,16], which archive the primitive diversity of the IFN system prior to further evolve in higher vertebrates, including humans [1,12]. Therefore, not only are amphibians unique for the study of the signature event of IFN evolution that leads to antiviral superiority of intronless IFNs in amniotes, but amphibian models are also irreplaceable in order to determine the evolutionary coordination and functional diversity among the IFN subgroups [12,13,14,15,16]. This is because amphibians represent the only animal clade to have both IFN gene subgroups (intronless vs. intron-containing) in either IFN-I or IFN-III types (Table 1 and Figure 1) [12,13,14,15,16].

## 2. The Fish-Like Intron-Containing IFN Genes in Amphibians

Two types of fish IFN genes, generally one to several IFN-γ and several IFN-ϕ (current nomenclature as fish IFN-a/-b/-c/-d, etc.) genes in each species have been previously only found in teleosts [10,11,12]. As fish IFN-γ manifests orthologous phylogeny with the IFN-II identified in tetrapods, it is uncertain about IFN-ϕ genes, which were proposed to be ancestral molecules for either IFN-I or IFN-III, or both, found in tetrapods. Per gene structure, all fish IFN-ϕ genes contain about four introns, which resemble what have been observed in most IFN-III genes but not IFN-I genes in amniotes, where most IFN-I genes are intronless [10,11,12,46,47,48,49]. Examination of the proteins coded by fish IFN-ϕ genes indicated that they contain either two or four cysteines to form into intra-molecular disulfide bonds critical for IFN tertiary conformations and potential antiviral potency [10,11,12]. A recent study identified an ancestral IFN-III gene in searching transcriptome of catshark, a cartilaginous fish [12]. This resolves that the jawed vertebrate ancestor has orthologs of all three interferon types [12]. Therefore, the IFN-ϕ genes previously discovered in bony fish, including both two- and four-cysteine groups, are determined to be in the IFN-I lineage, even though they may not be orthologous to tetrapod intronless IFN-I genes [46,47,48,49].

Amphibians bridge vertebrate evolution from the water to terrestrial lives. From the perspective of IFN evolution, previous studies only found several intron-containing IFN genes belonging to IFN types, which thus assumed more ancestral conservation than rapid IFN evolution in amphibians [9,11]. In spite of this, extensive inspection of current *Xenopus* genome assemblies identified 6–9 intron-containing IFN genes in both IFN-I and IFN-III types, which indicated a moderate gene expansion to compare to that identified in most fish species (Figure 1 and Table 1) [9,12,13]. Most amphibian intron-containing IFN genes encode a protein with four cysteines, with some exceptions of two or three cysteines in both IFN-I and IFN-III types [9,12,13,16]. Phylogenically, *Xenopus* intron-containing IFN-I genes belong to two groups (Group 4s in Figure 1): One (consisting of only three IFN genes) falls sister to fish ancestral and tetrapod IFN-I genes, and another group sisters amphibian intronless IFN-I genes (Figure 1). This indicates that amphibian intron-containing IFN genes might experience less birth–death rates than their amniotic intronless counterparts to preserve potential orthologs that derived into IFN lineages discovered in tetrapod [12]. In addition, these primitive amphibian IFN genes also adapted to evolve into different subgroups of intronless IFN genes, which believably have more rapid birth–death dynamics to evolutionarily cope with pathogenic and physiological pressure during the amphibiotic stage of vertebrate evolution [11,12].

## 3. The Expansion of Amniote-Like Intronless IFN Genes in Amphibians

In addition to the conservation of intron-containing IFN genes, the co-existence of intronless IFN genes inscribes the amphibiotic role of amphibians in vertebrate evolution [12,13,16,50,51]. Prior to identification of intronless IFN genes in amphibians, a prevalent model for IFN evolution described that a retroposition process happened in reptiles and led to the origin and expansion of intronless IFN-I genes in amniotes [9,10]. The discovery of intronless IFN genes in amphibians revised this model and confirmed that the emergence of intronless IFN might happen in amphibians or even earlier. Indeed, the earliest existence of intronless IFN genes so far was only detected independently in different species of amphibians [12,13,16,50,51]. Genes without introns are a characteristic feature of prokaryotes; however, a small portion of intronless genes also exist (even emerge) in animal genomes, including all mammalian species. In eukaryotic cells, genes use introns to enhance transcription fidelity, i.e., intron–exon splicing inhibits topoisomerase-I cutting activity to reduce mutagenesis generated during gene transcription [52]. On the other hand, intronless genes express more rapidly upon stimuli, and duplicate or mutate more efficiently for new gene evolution [50,52]. In this line, intronless IFN gene evolution may represent a prokaryote-wise arm race adapted by animal immunity at a molecular level [50].

Amniotic IFN-I genes are almost all intronless, which have been classified into multiple subtypes, including IFN-α, IFN-β, IFN-δ, IFN-ε, IFN-κ, IFN-τ, IFN-ω, and IFN-ζ, commonly (IFN-α, IFN-β, IFN-ε, and IFN-κ) or species-specifically (IFN-δ, IFN-τ, IFN-ω, and IFN-ζ) expressed in mammalians [11,12,13,14,15,16]. Birds and reptiles also possess several to a dozen intronless IFN-I genes in each species, which have also been named such as IFN-α and IFN-β, however, are not necessarily orthologs (but paralogs) to their mammal counterparts [11,12]. Amphibians contain intronless IFN genes of both IFN-I and IFN-III types. This is unique among all studied vertebrate species because amniotic IFN-I genes are generally intronless, and intron-containing IFN genes are common for all fish IFN genes and amniotic IFN-III genes [11,12,13,14,15,16]. In the three studied amphibian species (i.e., *Xenopus laevis*, *Xenopus tropicalis*, and *Nanorana parkeri*), each species may harbor several to near 40 intronless IFN genes, and the majority of them belonging to IFN-I types [12,13,14,16,50,51]. Intronless *Xenopus* IFN-I genes were grouped into four groups based on their intermolecular sequence similarity, and further defined into three clades based on their phylogenic relationship to other tetrapod species [12,13]. For example, about 10 intronless *Xenopus* IFN-I genes were located within the mammal and reptile IFN-I clade, next to a clade containing only reptile sequences [12,13]. This suggests that amphibians contain orthologs of amniote intronless IFN-I genes, and thus provide the earliest intronless ancestors (identified so far) of tetrapod IFN-I genes [12,13]. The other intronless amphibian IFN-I genes were nested or fell sister to fish intron-containing IFN genes, indicating their close phylogeny to these primitive ancestors as well [12]. Intriguingly, several *Xenopus* IFN-I genes were found to contain a single short intron and were assumed to be intermediate molecules recoding the intron loss process [13]. One of these small-intron containing *Xenopus* IFN-I genes located sister to the group of cartilaginous fish IFN-I genes that were next to the intronless amniote IFN-Is in a recent phylogenic analysis [12]. Thus, the intronless amphibian IFN-I genes consist of a complex of tetrapod IFN intermediates between fish intron-containing ancestors and amniotic IFN-I genes; however, a one-to-one orthologous relationship is yet to be determined between amphibian and amniotic intronless IFN genes [12,13].

In addition, amphibians are among the few tetrapod species having intronless IFN-III genes [12,13,51]. Studies indicated that intronless IFN-III genes have evolved at least twice during amphibian evolution [12]. We also detected the co-existence of intronless IFN-III genes in many amniotic species, and hence, at least three times intron loss has happened throughout vertebrate evolution [14]. However, intronless IFN-III genes experienced much less expansion potency than intronless IFN-I genes in tetrapods. It is unclear what kind of molecular or phylogenic mechanisms underlie this difference between IFN-I and IFN-III genes (Table 1 and Figure 2) [11,12,13,14,15,16].

## 4. Phylogenic View and Evolutionary Postulation about Amphibian IFN Evolution

In Figure 2, we summarize the current understanding of IFN evolution to highlight several key points and the newly discovered IFN complex in amphibians [12,13,14,15,16,50,51]. First, the ancestral molecules of three types of IFNs were identified in cartilaginous fish, indicating the earlier ramification of IFN primitive genes in fish [12]. This elegant discovery concludes a dispute about whether the fish IFN-ϕ genes are ancestral to tetrapod IFN-Is or IFN-IIIs to indicate that they all belong to IFN-Is, except a recently found SCCA-L from catshark, which is a fish IFN-III gene [12]. A question remains: Is that why no other IFN-III genes have been detected in bony fish so far? Extinct after the emergence or originated in a species- or taxon-independent manner? Second, intronless IFN genes in both IFN-I and IFN-III genes co-exist with intron-containing IFN genes, and this extensive co-existence has only been detected in different species of amphibians [12,13,16,50,51]. This indicates that the retroposition events that led to the emergence of intronless IFN genes happened several times in a species-independent way in amphibians [12,13,16,51]. Therefore, amphibians also comprise a unique animal taxon, containing both intronless and intron-containing IFN genes in both IFN-I and IFN-III types. Notably, intronless IFN-I genes but not IFN-III genes have experienced a rapid gene expansion, probably through a recently proposed birth–death process in *Xenopus* [12,13,16,51]. We propose that the emergence and expansion of intronless amphibian IFNs likely reflect the increase evolution pressure (especially the air-borne pathogens and physiological requirement for terrestrial adaptation) during the transition period when vertebrates migrated from aquatic to terrestrial environments. So why did intronless gene expansion happened in *Xenopus* but not obviously in *Nanorana* [13,16,51]? Maybe habitats close to human or domestic areas provides an epidemiological explanation [14,15]. Compared to intronless IFN-I genes, one paradox in IFN evolution is why IFN-II and –III genes conserve the ancestral gene structures, and do not show gene expansion even after few intronless IFN-III genes originated in amphibians and other ammonites. Recent studies discovered the epithelia-specified expression pattern and less-inflammatory activity of IFN-IIIs in mammalian studies [53,54,55,56,57]. It deserves investigation of whether this anatomic or physiological ramification limits the expansion potency of intronless IFN-III genes in tetrapods.

## 5. Amphibian IFN Complex: Expression and Function in Antiviral and Other Immune Responses

Described first for “interfering” with influenza virus in cultured chicken cells [58], IFNs are key antiviral cytokines and have multiple functions in immune regulation [4,5,6,7]. Nevertheless, most IFN functional studies have been conducted using mammal models, and amphibian IFNs remain largely uncharacterized. Previously, only several intron-containing IFN-I and IFN-III genes were identified in amphibians; and *Xenopus* IFN-III genes were ubiquitously detected in multiple tissues from healthy adult frogs [9,59,60]. *Xenopus* tissues or cells treated with poly (I:C) or infected with Ranavirus (FV3) robustly stimulated the expression of both types of IFN genes and their corresponding receptors. In *Xenopus* kidney cells and tadpole models, administration of recombinant IFN peptides of either *Xenopus* IFN1 or an IFNλ induced the expression of ISGs, including IFIT5, PKR, and MX1 [9,59,60]. The recombinant *Xenopus* IFN1 peptides protected the kidney-derived A6 cell line and tadpoles against FV3 infection, showing the decrease of the infectious viral burdens in both cases [59,60]. The study also observed that adult frogs were naturally more resistant to FV3 and were able to clear the infection within a few weeks, whereas tadpoles typically could not get over the viral infections. Correspondingly, adult *X. laevis* frogs demonstrated significantly higher FV3-elicited IFN gene expression than tadpoles, indicating that IFN response at least partly determined the antiviral consequence [59,60]. Even though IFN treatment markedly impaired viral replication and viral burdens in tadpoles, it only transiently increased tadpole survival time and did not change the final mortality rate of infected tadpoles due to considerable organ damage, regardless of IFN treatment in FV3-infected tadpoles [59,60]. Therefore, these studies revealed that amphibian intron-containing IFNs may consist of a basic immune barrier, even in healthy frogs; however, the undeveloped or irresponsive IFN status may contribute to the vulnerable consequence in tadpoles, which ascribe to a major portion of animal death in amphibian decline [59,60,61,62].

Genome-wide identification of intron-containing and intronless IFN genes in *Xenopus* and *Nanorana* species allows characterizing amphibian IFNs more extensively, in particular for the intronless subgroups [13,16]. The constitutive expression of multiple intronless IFN genes in *X. laevis* was demonstrated in most tested tissues, with a particular high expression in lung and heart [13]. The two intronless IFN-III genes were expressed in skin, stomach, and kidney. Similarly, the intron-containing IFN4 had the highest expression in kidney and heart. In contrast, IFN genes (such as the intronless IFNX3 and IFNLX group) that showed high inductive response (see context next) generally had a low expression level in normal tissues [13,50]. In addition to our gene-targeting expression analysis, most intronless IFN genes in *X. tropicalis* genome were also heavily mapped by RNA-Seq reads deposited at the public domains of NCBI, indicating functional expression of these less-studied amphibian IFNX genes (Table 2) [13].

Similarly, in the *N. parkeri*, an intron-containing IFN-I gene was highly expressed in kidney and intestine, as well as two intronless IFN-I genes in the lung, intestine, and skin, constitutively [16]. More than a 100-fold stimulation of the two intronless IFN genes was observed post-treatment of tissues (spleen, kidney, and liver) using poly(I:C) for 3–24 h [16]. More extensive inductive expression of amphibian IFN genes was compared in a kidney cell line from *X. laevis* (A6) after treatment of the cells with representative microbial mimics [13]. For example, intron-containing IFNs (represented by IFN4) were constitutively expressed at 5 h, but clearly showed inductive expression after cells were treated for 24 h. The viral dsRNA mimic, poly (I:C), most effectively stimulated several subgroups of IFNs, including both intron-containing and intronless IFNs in both IFN types at either 5 h or 24 h post-treatment. On the other hand, TLR2 and TLR4 ligands (Pam2/3CSK4/LTA and LPS, respectively) induced these same IFN expressions especially at 24 h post-treatment. Treatments with the ligands for TLR2/6 (Pam2CSK4) and TLR4 (LPS), but not ligands for TLR2/1 (Pam3CSK4) or TLR2 only (LTA), stimulated significant IFN expression. This indicates the differential regulation among amphibian IFN subgroups regarding defense responses against different pathogens [13].

The results from *Xenopus* tadpoles and adult frogs were more conclusive toward biological significance [61,62]. Using primary skin and kidney cells, Wendel et al. showed that most IFN genes were differentially expressed in the two cell types treated with various pathogenic mimics [62]. The findings include: (1) With few exceptions, most intron-containing IFN-I genes were more inductive in tadpole skin cells; (2) most intronless IFN-I and all IFN-III genes were more inductive in adult skin cells; and (3) the majority of IFN genes showed stronger induction in tadpole kidney cells; however, CpG DNA stimulated more robust IFN response in the adult kidney cells (Table 2). Post-FV3 infection, the expression of nine representative IFN genes (including six IFN-I and three IFN-III of both intron-containing and intronless groups) were increased 3–25-fold in the FV3-infected skin cells from both tadpoles and adult frogs; however, a robust stimulation (10–400-fold increase) of these IFN genes was detected in FV3-infected kidney cells from adult frogs, whereas negligible increases were detected in FV3-infected tadpole kidney cells [61,62]. Not only susceptible to amphibian FV3 infections, our data showed that frog kidney cells were also permissive to an influenza virus (H1N1 TX98) [13]. The flu virus infection induced a moderate increase of IFN4 and IFN-III gene expression at a low virus titer (multiplicity of infection or MOI at 1). A higher than 100-fold increase of these virus-responsive IFNs was detected in cells infected with the virus at a high MOI for 48 h. In addition, a strong expression of most intronless type I IFNs was shown in cells infected with the virus at 10 MOI, indicating a robust IFN reaction upon a massive viral infection [13]. In summary, the expression of these amphibian IFNs, particularly the intronless subgroup, was experimentally validated and shows differential expression in the ex vivo stimulated cells and animal tissues. Both the constitutive and inductive expressed patterns under various stimuli indicate the functional diversity of amphibian IFN complex as components of the innate immune system [13,16,59,60,61,62].

The antiviral activity of amphibian IFN peptides has been tested in both cells and animals [13,16,59,60,61,62]. Grayfer et al. produced a recombinant *Xenopus* IFN1 peptide using an insect cell-expression system [59,60]. The IFN1 peptide actively protected frog kidney cells against FV3 infection, and strongly induced MX1 (a robust antiviral ISG) expression in multiple tissues after administration into tadpoles for 24 h. In animal tests, the recombinant IFN1 peptide reduced viral burdens; however, did not improve the final mortality rate caused by FV3 infection in tadpoles [59,60]. Gan et al. tested the antiviral activity of *N. parkeri* intronless IFNi1 and IFNi2 by direct transfection of the IFN genes in a mammalian-expression vector into *Xenopus* kidney cells (A6 cells) [16]. They demonstrated that the transfection of IFNi1 and IFNi2 constructs induced the expression of serval typical ISGs and protected the cells against FV3 infection in the *Xenopus* kidney cells only if the IFNAR receptor genes of *N. parkeri* were co-transfected [16]. This indicates that there is a cross-species barrier of IFN antiviral activity due to species-specific IFN–IFNAR engagement in amphibians, further supporting the species-specific co-evolution of amphibian intronless IFN genes and corresponding IFN receptors [11,12,13,14,15,16]. We also used a mammalian-expression system to produce 15 *Xenopus* IFN peptides and determined their different antiviral activity against an influenza virus [13,14,15]. Antiviral assays by direct transfection of A6 cells using IFN-expressing constructs indicated that typical candidates of each amphibian IFN subgroup induced antiviral protection and reduced virus titers in A6 cells. In conclusion, the tested intron-containing IFN group (including IFN4, IFN6, and IFN7), which had higher activity against the viral cytopathic effect, are generally less active in suppression of viral replication. In contrast, the intronless IFNs demonstrated higher virus suppression, but were less effective to protect cell death from the viral infection [13,14]. Comparatively speaking, most *Xenopus* type III IFNs were less active in antiviral activity, except an IFNL4 that showed stronger virus-suppressive activity than most type I IFNs tested. Another significant IFN example was IFNX4, which demonstrated remarkable virus-suppressive and cell protective activity similar to most intron-containing IFNs tested. Clearly, all of these antiviral analyses validate that the *Xenopus* IFN complex comprises a profound antiviral diversity considering their antiviral activity against the influenza and FV3 infections [13,16,59,60,61,62].

The immune role of amphibian IFNs against other intracellular pathogens has also been recently tested [13]. *Listeria monocytogenes* represents a prevalent intracellular bacterial pathogen such as in food safety and public health. We detected that in the bacteria-infected A6 cells, most amphibian IFNs were highly stimulated when the cells were intracellularly infected with the bacteria. In all tests where cells were with different bacterial dosages, the intronless IFNX3 group was highly stimulated. This indicated that this intracellular IFN subtype (lack of signal peptides for secretion) might have evolved to regulate antibacterial activity inside cells. In cell cultures with a gel-overlay to restrict the bacterial intracellular growth, most IFN-expressing constructs showed antibacterial to some extent. However, most IFNs lost their antibacterial activity without a gel-overlay to limit the bacterial extracellular spreading. We also noted that IFNX2, the intracellular IFN isoform, developed significant antibacterial activity in both test conditions (with or without gel-overlay). Indeed, both intronless IFNX2 and IFNX3 are among the IFNX3 group, which showed the most antibacterial activity and were among the most inductive IFN genes in cells infected with *L. monocytogenes*, or treated with the ligands for either TLR2 (Pam2CSK4) or TLR4 (LPS) [13]. Considering that IFNX2 and IFNX3 are intracellular IFN peptides that lack a signal peptide for secretion, they potentially induce protective activity against intracellular bacteria via intracellular signaling pathways that are different from canonic IFN signaling through binding the membrane-bound IFNAR receptors [13,14,63]. Interferons and some ISGs (such as numerous IFN induced very large GTPase gene analogs) were found to be upregulated throughout Chytridiomycosis, a fungal skin infection that caused amphibian declines in several amphibian species [34,64]. Currently, there is lack of studies about the antifungal effects of amphibian IFNs. Several recent studies revealed a coordination between mouse type I IFNs and type III IFNs to regulate antifungal immune responses. Showing that during a fungal infection caused by *Aspergillus fumigatus*, type I IFNs produced by monocytes signaled the epithelial production of type III IFNs, which in turn played a critical role in regulating the antifungal neutrophil responses [65,66,67]. Given the diverse IFN-III genes evolved in amphibians, we expect to determine some amphibian IFN-IIIs that are important in antifungal immune response. In summary, recent studies conclude that the amphibian IFN family has a differential expression pattern and exerts a regulatory role in antiviral, antibacterial, and potential antifungal immune responses [13,16,59,60,61,62]. In addition, studies in amniotic IFNs also determined the multifunctional properties of antiviral IFNs in the regulation of immune cell development and metabolism, as well as animal pregnancy [68,69,70,71]. Whether amphibian IFN complex evolves to have similar functional diversification warrants further investigations in vertebrate physiology and development.

## 6. Conclusive Remarks: Amphibian IFN Complex—A Molecular Signature of Vertebrate Immune Evolution

Several recent studies have revealed a previously unknown complexity of the amphibian IFN system [11,12,13,14,15,16]. Being unique to animals from any other classes of vertebrates, amphibians not only conserve and multiply the fish-like intron-containing IFN genes, but also rapidly evolve amniote-like intronless IFN genes in three tested species [13,16,51]. Amphibians are thus critical intermediate species in IFN (and potentially other immune gene families too) evolution, in addition to their key position in studies of vertebrate development [24,35,40]. We postulate that the amphibian IFN system provides an irreplaceable model to study vertebrate immune evolution in molecular and functional diversity to cope with unprecedented immuno-physiological requirement during terrestrial adaption [11,12,13,14,15,16]. Studies so far have ascribed a defense role of these IFNs in immune regulation against intracellular pathogens particularly viruses and bacteria [13,16,59,60,61,62]; however, many knowledge gaps remain to be completed. Based on recent reports about IFN’s multifunctional properties in regulation of animal physiological and defense responses, we interpret that amphibian IFNs may evolve novel function pertinent to their superior molecular diversity [4,5,6,7,11,12,13,14,15,16]. Such new function revealed by the emerging studies of antifungal and developmental regulation of amphibian IFNs will certainly promote our understanding of immune evolution in vertebrates to address current pathogenic threats causing amphibian decline [25,26,27,28,29]. *Xenopus* is proposed as an alternative model for cancer studies due to its evolutionary tumor immunity [35]. Antitumor activity constitutes another key function of the antiviral IFNs [6]. We therefore expect that the amphibian IFN complex may have evolved IFN molecules bearing superior antitumor activity, and the *Xenopus* tumor immunity model can contribute to the development of IFN-based cancer immunotherapies [6,35].

## Figures and Tables

**Figure 1 cells-09-00067-f001:**
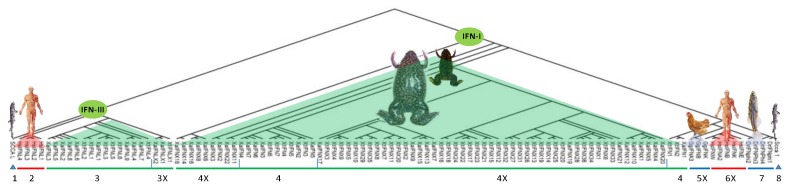
Schematic of vertebrate interferon (IFN) evolution and recent discovery about amphibian IFN complex. Evolutionary analyses were conducted in MEGA X [17]. The minimum evolution (ME) method was used to infer the molecular phylogeny, and the evolutionary distances were computed using the p-distance method, shown in the units of the number of amino acid differences per site. The ME tree was searched (search level of 1) using the close-neighbor-interchange (CNI) algorithm. The analysis used 108 amino acid sequences. All ambiguous positions were removed for each sequence pair. There was a total of 437 positions in the final dataset. In contrast to the previously known several fish-like intron-containing amphibian IFNs, recent studies revealed that amphibians are unique to have both fish-like intron-containing and amniotic intronless IFNs, which molecularly and immunologically stamp their amphibiotic position in vertebrate evolution [12,13,16]. Legends and Abbreviations: IFN-I, type I IFNs (including amniotic intronless IFNA, IFNB, IFNE, IFNK, IFNW, respectively, for the genes of IFN-α, -β, -ε, -κ, -ω; and diverse amphibian intron-containing, XaIFN or XtIFN, and intronless XaIFNX or XtIFNX, listed here); IFN-III, type III IFNs (including amniotic intron-containing IFNL gene for IFN-λ; and amphibian intron-containing XaIFNL or XtIFNL, and intronless XaIFNLX or XtIFNLX, listed here); SCCA 1, *Scyliorhinus canicula* (catshark) ancestral IFN-I; SCCA L, *Scyliorhinus canicula* (catshark) ancestral IFN-III; Dr, *Danio rerio* (zebrafish); Gg, *Gallus gallus* (chicken); Hs, *Homo sapiens*; Xa, *Xenopus laevis*; Xt, *Xenopus tropicalis*; Group designation: 1 and 7 & 8, intron-containing ancestral IFN-I (7 & 8) and IFN-III (1); 2, intron-containing IFN-III in amniotes; 3 and 4, amphibian intron-containing IFN-III (3) and IFN-I(4); 3X and 4X, intronless amphibian IFN-III (3X) and IFN-I (4X); 5X and 6X, intronless IFN-I in amniotes.

**Figure 2 cells-09-00067-f002:**
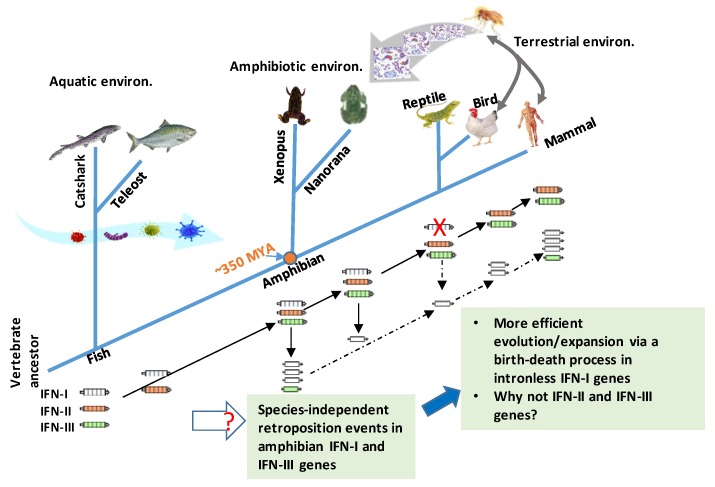
An updated model for antiviral IFN (IFN-I and IFN-III) gene evolution in vertebrates. The ancestral genes of three type of IFNs (IFN-I, -II, and -III) have been identified in cartilaginous fish. All fish IFN genes are intron-containing ones generally with four introns (or five exons) [12]. In contrast, amniotic IFN-I genes are generally intronless. Co-existence of multiple intronless and intron-containing IFN genes were only detected independently in different species of amphibians [12,13,14,15,16,50,51]. The current evidences support that at least two retroposition events might have happened in amphibians, at about 180.70 MYA (million years ago) in *Xenopus* and 87.57 MYA in *N. parkeri*, for example [12,13,14,15,16,50,51]. These two independent retroposition events might have occurred much later than the divergence between amphibians and amniotes (i.e., ~350 MYA, orange dot). Another retroposition event leading to intronless IFN orthologs in amniotes could also happen in accompaniment to loss of intron-containing IFN genes in reptiles; however, it could be unlikely if an orthologous relationship exists between different intronless amphibian IFNs with amniotic ones [12]. The emergence and expansion of intronless amphibian IFNs likely reflect the increasing evolution pressure (especially the air-borne pathogens and physiological requirement for terrestrial adaptation, shown with broad curve arrows) during the transition period, when vertebrates migrated from aquatic to terrestrial environments. Retroposition defines a reverse-transcription process of cellular mRNA and reintegration into the genome to enhance gene copying and evolution in molecular evolution. Compared to intronless IFN-I genes, one paradox in IFN evolution is why IFN-II and –III genes conserve the ancestral gene structures, and do not show gene expansion even after few intronless IFN-III genes originated in amphibians and other amniotes. Solid or dashed black arrows: Indicating certain or uncertain orthologous relationships discovered, respectively.

**Table 1 cells-09-00067-t001:** Molecular composition of amphibian IFN complex compared with IFNs in amniotes (human) and fish (catshark).

IFN Type	Gene/Subtype *	Fish: Catshark	Amniotes: Human	Amphibian	
				(*X. tropicalis*	*X. laevis*)
IFN-I	Intron-containing Intronless	1 0	0 17/5	7/2 36/4	7/2 22/5
IFN-II	Intron-containing	1	1	1	1
IFN-III	Intron-containing Intronless	1 0	4 0	6 1	9 2

* Gene number/subtype number. Subtype classifications are based on both molecular phylogeny and function for amniotic IFNs (human), but primarily molecular phylogeny in amphibian IFNs [12,13].

**Table 2 cells-09-00067-t002:** Functional knowns and unknowns about amphibian IFN complex referring to well-studied amniotic IFNs.

IFN Type	Function Properties [1,2,3,4,5,6,7]	Amniote: Human and Mouse [1,2,3,4,5,6,7,53,54,55,56,57,58,59]	Amphibian [13,16,59,60,61,62]
**IFN-I**	**Principal producer cells**	Basically all nucleated cells, but major producers: Leukocytes (IFN-α), fibroblasts (IFN-β), macrophages/pDCs (IFN-α), epithelial cells (IFN-ε/-κ)	Major tested cells: Bone marrow-derived macrophages, skin cells, and kidney cells
	**Inducing agents**	Viruses, dsRNA, CpG, IFNs	Viruses, dsRNA, ssRNA, CpG, IFNs(?)
	**Classical activity:**		
	Engaging IFNAR1/2 to induce ISGs Antiviral Antiproliferation Antitumor Immunomodulation	Yes, direct evidence from IFN-α/-β/-ω subtypes Yes, many viruses in vitro/vivo Differentially of subtypes in various cell types Yes, in cancer/tumor cells Regulation of inflammation, B-cell and T-cell generation and activation	Probably, IFNAR1/2 and most ISGs are there, limited evidence Yes, tested on Flu and FV3 Unknown Unknown Unknown
	**Unclassical function:**		
	Developmental and tissue-specific regulation Reproduction signaling Non-canonical signaling	IFN-β: Microbiota; IFN-α: Embryonic HSC; IFN-ε: Reproductive tract; IFN-κ: Keratinocytes IFN-ω/τ (in pigs and cattle) Potentially via IFNAR2 only, AMPK/mTOR, etc.	Unknown Unknown Unknown
**IFN-II**	**Principal producer cells**	Activated T cells, NK cells, macrophages	No direct evidence, maybe similar to amniotic counterpart
	**Inducing agents**	Mitogens, antigens, and IL-2	LPS, dsRNA, mitogens/antigens/IL-2(?)
	**Major activity**	Immune regulation on leukocytes; antimicrobial, antitumor	Incomplete evidence, maybe similar to amniotic counterpart
**IFN-III**	**Principal producer cells**	Epithelial cells, BMCs, blood cells	Skin cells, kidney cells, and macrophages
	**Inducing agents**	Similar to agents for IFN-I	Similar to agents for IFN-I
	**Major activity:**		
	Engaging IFNLR1/2 to induce ISGs Epithelia-specific antiviral Anti-inflammatory effect Antifungal	Direct evidence from IFN-λ1/λ3 Especially for gut and respiratory tract Yes, prone to functional protection on mucosa Yes, act through activation of neutrophils	No direct evidence, maybe similar to amniotic counterpart May not limit to epithelia (?) Unknown Potential, incomplete evidence

**Acronyms:** AMPK, AMP-activated protein kinase; BMC, bone marrow cells; CpG, synthetic DNA motif of TLR9 ligand; dsRNA, synthetic double-strand RNA; FV3, frog virus 3; HSC, hematopoietic stem cell; IFN, interferon; IFNAR, IFN-I receptor; IFNLR, IFN-III receptor; IL, interleukin; ISG, IFN-stimulating genes; LPS, Lipopolysaccharide; mTOR, mammalian target of Rapamycin; NK cell, nature killer cell; ssRNA, single-strand RNA.

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
