# Peer review of "Xenopus Interferon Complex: Inscribing the Amphibiotic Adaption and Species-Specific Pathogenic Pressure in Vertebrate Evolution?"

_cells, 2019, doi:10.3390/cells9010067_

Round 1

Reviewer 1 Report

The paper from Tian et coll. is sounding and focused on the evolution of the immune response in amphibians.

The authors have extensively reviewed the literature and prepared a complete, fluent paper that explores in depth the fundamentals of the immunobiology.

I have really appreciated this paper, and only suggest a review to eliminate some errors in the text.

Reviewer 2 Report

This is an interesting and an informative review on the evolution of amphibian type I and type III IFNs.

Please find my suggestions below:

Line 35: IFN phi-like genes, which are presumably to be progenitors and evolve into type II IFNs. Please change this sentence for grammatical and scientific reasons. The teleost IFN-phi are the fish intron-containing type I IFNs not type II IFNs.

Again in Line 129, the authors should be aware that many bony fish species have multiple type II IFNs, and the current nomenclature for type I IFNs is IFNa, IFNb, etc. It would be pertinent for the authors to expand on this notion, relating to number of cysteines (mentioned) and function (some are more antiviral, others have no observed effects) as this may help explain the expansion and possible neo-functionalization of the amphibian IFNs

Please also add teleost fish (maybe a cyprinid, that has multiple type II IFNs and a salmonid, which has single type II but well characterized type I IFNs) to table 1

The authors need to address the numerous type I IFN type in fish (IFNas, bs, cs, ds, etc) which were previously called IFN phi.

The authors also need to address the fact that some bony fish have multiple type II IFNs and IFN receptors, with splice variants and isoforms of IFNg  (IFNg2) and IFNg-related (IFNgrel, IFNg1).

Amphibia represent a very large and diverse kingdom of animals with distinct physiologies and immune systems; consisting of anurans (which themselves are very immunologically dispirate), Caudata, Anura and Apoda. Please be very precise on what information is available on which groups and species of amphibians with regard to type I and type III IFNs. In absence of information on many of these species, you may consider changing the title to reflect the species in which this information is known. Please also provide a table list of the Xenopus and Nanorana IFNs, highlighting similarities and differences.

Table 2. Please note that amphibian (X. laevis) macrophage populations have been described as major (not low level) producers of IFNs I and III. Please correct this accordingly.

Reviewer 3 Report

This review about the newly identified IFN molecular evolution and expansion in amphibians shows a unique role of amphibians in IFN evolution. Indeed, amphibians are unique as they have both fish-like intron-containing and amniotic intron-less IFN genes. Thus, the amphibian IFN system provides an essential model to study vertebrate immune evolution in molecular and functional diversity to cope with unprecedented pathophysiological requirement during terrestrial adaption. It appears that the amphibian IFNs may have evolved novel function pertinent to their superior molecular diversity. Such new function could promote understanding of immune evolution in vertebrates to address current pathogenic threats causing the amphibian decline, a current major ecological problem.

This review is well written, comprehensive and nicely illustrated.

I only have two very minor comments:

- Lines 148-9, authors indicate that some amphibian intron-containing IFN genes encode a protein with three cysteines. Given the odd number of cysteines in these proteins, readers might wonder what is the function of the third cysteine? Is there some indication about its function?

- Table 1, to ease understanding by readers, please indicate in the legend of this table what means the ‘/’.
